# Novel Biomaterials in Glaucoma Treatment

**DOI:** 10.3390/biomedicines12040813

**Published:** 2024-04-07

**Authors:** Adéla Klézlová, Petr Bulíř, Andrea Klápšťová, Magdaléna Netuková, Kateřina Šenková, Jana Horáková, Pavel Studený

**Affiliations:** 1Ophthalmology Department, Third Faculty of Medicine, Charles University, University Hospital Kralovske Vinohrady, Ruská 87, Praha 10, 100 00 Prague, Czech Republic; petr.bulir@nemlib.cz (P.B.); magdalena.netukova@fnkv.cz (M.N.); katerina.senkova@fnkv.cz (K.Š.); pavel.studeny@fnkv.cz (P.S.); 2Department of Ophthalmology, Regional Hospital Liberec, 460 01 Liberec, Czech Republic; 3Department of Nonwovens and Nanofibrous Materials, Faculty of Textile Engineering, Technical University of Liberec, 461 17 Liberec, Czech Republic; andrea.klapstova@tul.cz (A.K.); jana.horakova@tul.cz (J.H.)

**Keywords:** glaucoma, biomaterials, glaucoma drainage implants, drug delivery systems

## Abstract

Glaucoma is a significant cause of blindness worldwide, and its treatment remains challenging. The disease progressively leads to damage to the optic disc and thus loss of visual acuity and visual field. High intraocular pressure (IOP) is a common risk factor. There are three major methods to treat this disease: topical, laser, and surgical. None of these are completely satisfactory; therefore, alternatives using new biomaterials are being sought. Since biomaterial engineering has experienced significant growth in recent decades, its products are gradually being introduced to various branches of medicine, with the exception of ophthalmology. Biomaterials, such as glaucoma drainage implants, have been successfully used to treat glaucoma. There is significant ongoing research on biomaterials as drug delivery systems that could overcome the disadvantages of topical glaucoma treatment, such as poor intraocular penetration or frequent drug administration. This article summarizes the use of novel biomaterials for glaucoma treatment presented in the literature. The literature search was based on articles published in English on PubMed.gov, Cochranelibrary.com, and Scopus.com between 2018 and 2023 using the following term “biomaterials in glaucoma.” A total of 103 published articles, including twenty-two reviews, were included. Fifty-nine articles were excluded on the basis of their titles and abstracts.

## 1. Introduction

In 2010, glaucoma caused blindness in approximately 2.1 million people worldwide. This eye disease is not only a problem in less-developed countries, but even in Western Europe it is the second most common cause of irreversible vision loss after age-related macular degeneration [1]. It leads to thinning of the retinal nerve fiber layers, causes progressive optic nerve damage, and causes gradual loss of sight in both adults and children [2]. Pharmaceutical and surgical treatment, focusing on the management of intraocular pressure (IOP), can significantly slow the adverse changes and prevent the irreversible loss of visual functions [3]. However, the current therapeutic methods are not completely effective or completely stable and have complications. Therefore, glaucoma treatment remains challenging, and research into new methods and materials that are more durable, long-standing, biocompatible, and effective in lowering IOP is being conducted.

Biomaterial engineering has experienced extensive growth in recent decades, and its products are being widely used in therapeutic and diagnostic medicine. Biomaterials have been implanted in tens of millions of patients worldwide, and over 13 million interventions occur annually in the United States [4]. The first biomaterials were designed to perform mechanical functions, for example, implants, which are used in a wide spectrum of medical industries. Due to their biocompatibility, mechanical strength, and resistance to corrosion, metals and alloys such as titanium are used as substitutes for heart valves, vessel stents, or joint implants [5].

The eye can be a more ideal environment for biomaterials than other body systems due to the following. The immune privilege status of the eye contributes to a limited immune response to these materials and, on top of that, the presence of the blood–retinal barrier limits the systemic penetration of most biomaterials and thereby reduces the risk for developing severe systemic side effects [6,7]. There are three major pathways for reducing IOP, the main risk factor. IOP is the only directly influential factor affecting the progression of glaucoma and can be influenced by conservative, laser, or surgical glaucoma treatments. All have advantages and disadvantages, side effects, and limitations. Biomaterials have already been successfully used; for example, in the surgical treatment of glaucoma, they are used as glaucoma drainage implants (GDIs). The first device, the Molteno implant (1969), was made of polypropylene. Since then, many other GDIs have been developed from different types of biomaterials, such as polypropylene, polyethylene, and gelatin. Nonetheless, all types of surgical glaucoma treatments show very little success in reducing IOP or improving long-term functionality. Long-term studies have shown that, even when IOP is sufficiently reduced, more than half of the patients still need additional treatment [8,9,10].

Currently, there is research into new biomaterial drug delivery systems, which may eliminate the drawbacks of the traditional topical application of anti-glaucoma drugs. Glaucoma is a chronic disease that often affects individuals for decades. It requires the daily use of anti-glaucoma drops, mostly in different combinations; as such, it often leads to compliance issues. Drug bioavailability and side effects should also be considered. New anti-glaucoma delivery systems using new biomaterials have great potential. 

In this review, we summarize the latest trends in the use of biomaterials for glaucoma treatment. There are many different materials, together with many different manufacturing procedures, that have been tested both in vitro and in vivo in recent years. Many of them have shown promising results; however, further investigation is necessary. We believe that advancements in nanofiber materials and their application to glaucoma could be a way to successfully treat the progression of this potentially blinding disease.

## 2. Method

Research in the current literature is based on the use of novel biomaterials for glaucoma treatment. An extensive search was created based on articles published in English on PubMed.gov, Cochranelibrary.com, and Scopus.com between 2018 and 2023 using the term “biomaterials in glaucoma.” A total of 103 published articles, including twenty-two reviews, were included. Fifty-nine articles that were unrelated to the use of biomaterials in glaucoma treatment were excluded. This article is divided into three main chapters: (1) biomaterials in drainage implants, (2) prevention of fibrosis in glaucoma filtration surgery, and (3) drug delivery systems (DDSs). 

## 3. Types of Biomaterials

Biomaterials can be divided according to their origin: natural, synthetic, and composite. Natural biological materials, usually natural polymers, have several valuable properties. They can support effective remodeling, do not usually cause foreign body responses, and do not lead to tissue scarring or encapsulation. However, their mechanical properties and architecture are subject to natural variability, are less cost-effective, and have poor reproducibility. Synthetic materials have highly tunable properties, and they can be easily and precisely adapted during manufacturing according to their planned usage. However, they do not promote constructive tissue remodeling and can be responsible for fibrosis and undesirable foreign body reactions. Composites of natural and synthetic materials combine the positive qualities of both materials [11].

### 3.1. Novel Biomaterials in the Surgical Treatment of Glaucoma

Glaucoma can be controlled by medications, laser surgeries, or filter surgeries, such as gold standard trabeculectomy [12]. Glaucoma drainage implants (GDIs) are currently one of the possible options for surgical therapy. Their function mechanism to reduce IOP is well described in the published literature [13,14]. These devices use the unconventional outflow of aqueous humor, which is drained from the anterior chamber into a reservoir associated with the implant, and the fluid is further reabsorbed by the body. Materials such as silicon and polypropylene are often used for the development of GDIs. The first GDI was the Molteno implant, which was developed in 1969 and was made of polypropylene. The relatively short time period of five years of functionality is described in half of all GDIs; therefore, intensive studies continue to improve the biomaterial, technique, and shape properties of GDIs [15]. An overview of new biomaterials used for glaucoma drainage implant development is presented in Table 1.

One of the newer GDIs approved by the U.S. Food and Drug Administration (FDA) in November 2016 is the XEN Gel Implant (Allergan INC, Dublin, Ireland). Detailed illustration of the implant can be found at https://hcp.xengelstent.com/ (accessed on 3 April 2024). XEN uses minimally invasive glaucoma surgery (MIGS) to reduce IOP without the need for extensive surgical procedure [16]. The implant is developed from collagen-derived, animal-based gelatin, which was chosen due to its promising properties such as flexibility, ability to chemically crosslink to form durable implants, and its known characteristic of not causing foreign body reactions. Furthermore, collagen is the main component of connective tissue. The next advantages of collagen are its well-documented structural, physical, chemical, and immunological properties, biodegradability, and biocompatibility. Its easy isolation and purification means that it can be produced in large quantities, which is an important criterion for further use [17].

In 2019, Denis et al. published the first in vivo study on the efficacy and safety of MINIject implants in patients with medically uncontrolled open-angle glaucoma. MINIject (iSTAR Medical SA, Wavre, Belgium) is an implant that facilitates the outflow of the aqueous humor using the pathway from the anterior chamber via the supraciliary space, thus reducing IOP. A detailed illustration of the implant can be found at https://www.istar-medical.com/treatments/ (accessed on 3 April 2024). It was developed from biocompatible STAR material, which is a soft and flexible medical-grade silicone with an innovative porous design that encourages a natural flow speed and should prevent the incidence of fibrosis and scarring, which is responsible for short implant effectiveness [18]. The MINIject™ micro-invasive glaucoma implant was also evaluated in a rabbit model by Grierson et al. in 2020. The authors concluded that the implant was well tolerated in the ocular tissues of rabbits, was bio-integrated, and showed good biocompatibility. Owing to minimal fibrous encapsulation, the drainage capacity of the implant was preserved over time [19].

A novel material for the treatment of glaucoma was described by Klapstova et al. in 2021 using in vitro experiments. The material was based on nontoxic, biocompatible, and non-degradable polyvinylidene fluoride (PVDF) developed using electrospinning technology. The authors used three different types of PVDF with different average molecular weights in the experiment and evaluated their therapeutic properties. PVDF was used in two forms: as a pure polymer and blended with polyethylene oxide (PEO). The experiment showed that the composite nanofibrous implant made of PVDF/PEO inhibited cell growth without the addition of antifibrotic agents. It prevented the cell fibrotization of 3T3 mouse fibroblasts and associated blockage of the system [20].

In 2021, Kao et al. examined a novel ultrathin tubeless subconjunctival shunt, VisiPlate, whose structure corresponds to a network of microchannels. The studied shunt was made of a 400 nm thick freestanding aluminum oxide (alumina) corrugated plate coated with a two µm thick layer of parylene-C. The authors evaluated its biocompatibility and ability to reduce IOP in rabbit eyes, and suggested that it could safely improve aqueous humor outflow and significantly reduce IOP [21].

In 2023, Josyula et al. published a subsequent study of their previously developed synthetic nanofiber-based GDIs made of non-degradable polyethylene terephthalate (PET) with partially degradable inner cores made from polyglycolide (PGA). The authors evaluated GDIs with nanofibers or smooth surfaces to examine the influence of surface topography on the implant capability. In addition, implants in this study were tested on rabbit eyes, where the experiments showed that GDIs with a nanofiber structure were biocompatible, did not cause hypotony, and ensured a similar outflow as the commercially available GDIs; additionally, a decreased tendency to fibrotic encapsulation and expression of fibrotic markers was described with these GDIs [22].

The development and evaluation of silicone elastomer-based microstents were described by Siewert et al. The safety and efficacy of fully processed, minimally invasive implanted microstents were analyzed in rabbits with a monitoring time of six months. The authors suggested that their microstent, with unique valve-controlled drug-releasing properties, may be a novel treatment option for the clinical management of glaucoma [23]. 

The importance of the type of biomaterial was demonstrated in a publication by Lubinski W., published in 2018, addressing the treatment of neovascular glaucoma. The authors compared the clinical outcomes two years after the implantation of polypropylene or silicone Ahmed^®^ glaucoma valves in patients with neovascular glaucoma. Both materials have a long tradition in glaucoma treatment, but the authors drew attention to the influence of different materials on visual outcomes after Ahmed^®^ valve implantation for neovascular glaucoma, most likely due to the differences in the biocompatibility of the materials [24].

In addition, magnesium alloys seem to be an interesting new biomaterial with good potential for the development of modern drainage devices for glaucoma surgical treatment [25]. A novel stent with hyaluronic acid hydrogel filling was also examined experimentally [26].

**Table 1 biomedicines-12-00813-t001:** Novel biomaterials used in glaucoma drainage implants (GDIs).

Type of GDI	Biomaterial	Author, Year	Type of Study
XEN Gel Implant	Animal-based gelatin	Chaudhary, 2016 [16]	In vivo
Magnesium alloy	Magnesium alloy	Li, 2018 [25]	In vitro
Hyaluronic acid hydrogel filling stent	Hyaluronic acid hydrogel	Thaller, 2018 [26]	In vitro
MINIject (iSTAR Medical SA)	Medical-grade silicone	Denis, 2019 [18]	In vivo
PVDF implant	Polyvinylidene fluoride	Klapstova, 2021 [20]	In vitro
VisiPlate	Aluminum oxide + paryleneC coating	Kao, 2021 [21]	In vivo
Nanofiber-based GDI	PET/PGA	Josyula, 2023 [22]	In vivo
Silicone elastomer-based microstent	Silicone elastomer-based	Siewert, 2023 [23]	In vitro + Preclinical In vivo

### 3.2. Novel Biomaterials in the Prevention of Fibrosis in Glaucoma Filtration Surgery

Conjunctival fibrosis remains a major impediment to the success of glaucoma filtration surgery, and new therapies and biomaterials for the mitigation of fibrosis are required. 

Shao et al. focused on new, potentially effective options for preventing fibrosis after glaucoma filtration surgery. The authors dedicate one of their chapters to new biomaterials, such as hydrogels with sustained release of antimetabolites, liposomal delivery systems, encapsulation of therapies in nanoparticles, etc. [27].

There are studies developing and evaluating drainage devices that can prevent postoperative fibrosis; for example, Zhao et al., 2021 evaluated a 5-fluorouracil (5-FU)-loaded chitosan microtube (CMT) for glaucoma aqueous humor drainage and Ioannou N et al. studied 3D-printed implants composed of polycaprolactone (PCL) and chitosan (CS) loaded with 1% 5-FU [28,29]. In another study, Dong et al. modified an Ahmed valve by absorbing MMC-loaded opal shale microparticles to prevent scar formation after glaucoma drainage device implantation surgery [30]. 

Chun YY et al. reported a biocompatible injectable gelatin-based hydrogel serving as a platform for siRNA (small interfering RNA) protection and delivery of anti-fibrotic treatment. The hydrogel platform was used to deliver siSPARC (siRNA for secreted protein, acidic and rich in cysteine) in in vivo experiments using a rabbit model and proved to be effective in reducing subconjunctival scarring after glaucoma filtration surgery [31].

### 3.3. Novel Biomaterials in Conservative Glaucoma Treatment

Another approach to reducing IOP is pharmacology-based methods. Topical drug delivery into the eye is the most popular and available treatment route for many eye diseases [32]. Topical eye drops, ointments, and oral medications are commonly used to decrease IOP; however, they have limitations, including poor patient compliance and inadequate bioavailability [33]. To increase bioavailability and improve patient compliance, advanced drug delivery mechanisms, such as liposomes, microneedles, niosomes, dendrimers, ocular inserts, nanoparticles, and injectable hydrogels have been described in the literature [34,35]. In situ hydrogels can potentially act as delivery vehicles for nutrients, oxygen, and drugs in a targeted area [36]. Hydrogels are created from natural biopolymers such as silk fibroin, chitosan, and alginate, and their features, such as biodegradability, biostability, biocompatibility, and non-toxicity, are typical for them. This offers great potential for the development of drug delivery systems for the management of a wide spectrum of ocular impairments, including glaucoma [37,38,39]. Nanoparticles are particles with a dimension of less than 100 nm and can possess a surface charge, depending on the monomer properties, which allows for a better permeability or mucoadhesion of therapeutics. Nanoparticles act through the encapsulation of the target therapeutic or surface loading through electrostatic interactions [40,41]. The advancement of materials science gives us a wide range of advanced biocompatible polymers with adjustable mechanical properties and degradation rates suitable for drug carriers [42]. One of the important attributes of the polymers used in DDSs is biodegradability and the related biodegradation rate, which is responsible for the controlled release of drugs [43,44]. The abundance of polymers available for drug delivery is vast, but they generally fall into two categories: synthetic polymers and biopolymers [45].

#### 3.3.1. Natural Biomaterials in Drug Delivery Systems (DDSs)

##### Chitosan-Based/Coated DDSs

Chitosan (CH) is a very popular bio-adhesive polymer that serves as the basis for the development of many ophthalmic DDSs. It is a linear, cationic polysaccharide with a natural origin and is produced by the alkaline deacetylation of chitin [46]. Various scientists have used CH as a coating material for the newly invented ocular DDSs [32,47,48].

El-Feky et al. tested a hydrogel made from CH-Gelatin and crosslinked it with oxidized sucrose as a possible ocular delivery system for timolol maleate to control ocular hypertension. The authors summarized, according to the results of their in vitro and in vivo studies, that the formulated hydrogel was able to maintain release rates and the effectiveness of timolol for a longer time than commonly used topical eye drops [49]. Pakzad et al. (2020) also concluded that timolol maleate-loaded quaternized CH-based thermosensitive hydrogel is a promising candidate for DDSs in glaucoma treatment [50].

Nguyen et al. described the role and importance of aromatic ring numbers in phenolic compound-conjugated CH injectables and explained their influence on the effectiveness of glaucoma treatment [51]. In another study, Nguyen et al. reported that the generation of amine-terminated polyamidoamine dendrimers in injectable thermogels was key to achieving extended drug release profiles and anti-inflammatory effects. These findings show great potential for the medical management of advanced glaucoma [52].

Li et al. (2018) described the use of montmorillonite/CH nanoparticles as a new controlled-release topical ocular delivery system for glaucoma therapy. The authors intercalated betaxolol hydrochloride (BH) into the interlayer gallery of natrium–montmorillonite and further enhanced CH nanoparticles. Different in vitro tests, such as release performance, analysis of irritation, and precorneal retention capability, were conducted, and the authors concluded that BH-Mt/CS NPs could represent a new class of nanocarriers, providing better compliance and therapeutic results in glaucoma patients [53].

CH-coated bovine serum albumin nanoparticles (BSA-NPs) for topical delivery of tetrandrine (TET) were developed and studied by Radwan et al. CH coating modulated drug release and improved the mucoadhesion of the NPs. Owing to the CH coat, the transcorneal penetration of the rabbit corneas was enhanced. CH-coated TET-BSA-NPs showed important improvements in the pharmacokinetic parameters of TET, and thus lowered IOP in rabbit eyes. In addition, the nanoparticles in this study appear to be a promising approach for more efficient glaucoma treatment [54].

Luo et al. introduced a strategy to develop nano-eye drops for the treatment of glaucoma progression. They functionalized CH and ZM241385 onto the surfaces of hollow ceria nanoparticles (hCe NPs); this dual-functional therapeutic platform for sustained intraocular delivery of pilocarpine was then assessed. The authors’ findings showed great promise for the development of novel nano-eye drops, offering effective therapy for ocular pathologies, including the inner parts of the eye, together with glaucoma [55]. The same authors also invented an injectable and biodegradable chitosan-g-poly(N-isopropyl acrylamide) (CN) material that could provide the continuous release of pilocarpine and decrease high IOP in glaucomatous eyes to near normal values for at least two months [56].

Mucoadhesive polymeric inserts prepared using CH and chondroitin sulfate were used as DDSs for benzamidine in the research of Cesar et al. This study showed promising results for anti-glaucoma treatment [57]. 

CH/hydroxyethyl cellulose-based ocular inserts for the sustained release of dorzolamide were developed and evaluated in a study by Franca in 2019. The efficiency of the inserts was tested in glaucomatous rats. The findings of the authors showed encouraging results for the future use of polymeric-based inserts for the sustained release of dorzolamide in glaucoma [58].

CH-pectin mucoadhesive nanocapsules (CPNCs) loaded with brinzolamide were prepared, and their efficacy in glaucoma treatment was evaluated in a study by Dubey et al. Pharmacodynamic studies were conducted for CPNCs in the glaucoma-induced rabbit eye model and compared with the available commonly used eye medications. The authors concluded that these nanocapsules were a reasonable choice over conventional eye drops due to their improved bioavailability owing to their longer precorneal time and their ability to release the drug continuously [59].

Niosomes (NIM) of carteolol (CT) were developed using the thin-film hydration method and were optimized using the Box–Behnken statistical design described by Zafar et al. (2021). The optimized CT-NIM was coated with CH to prolong the time of action in the eye. The authors concluded that NIM could be a potential carrier for the delivery of CT with a better intraocular effective time [60].

Li et al. (2020) compared three types of CH nanoparticles. Carboxymethyl CH (CMC), hydroxypropyl CH (HPC), and trimethyl CH (TMC) were used as cationic materials to prepare tetrandrine lipid nanoparticles (TET-LNPs) for glaucoma therapy. The monitored parameters included in vitro drug release, precorneal retention, ocular irritation, and drug biofilm interactions. In this study, the evaluated CH nanoparticles showed good properties for ocular administration [61].

##### Montmorillonite-Based/Loaded DDSs

Montmorillonite-loaded solid lipid nanoparticles using BH were tested in a study by Liu et al. and their promising use in ophthalmic delivery systems was described [62].

Finally, the montmorillonite/betaxolol hydrochloride complex encapsulated into Eudragit microspheres was studied by Tian et al. with promising in vitro effects. Eudragit RS and RL polymers are often used for research on controlled-release drug forms. They have a positive charge, which prolongs the residence time on the corneal surface [63,64].

A novel micro-interactive dual-functioning sustained-release delivery system (MIDFDS) for glaucoma treatment was described in 2021 by Liu et al. Betaxolol hydrochloride molecules were incorporated into montmorillonite by ion exchange, and MIDFDS formation was confirmed by X-ray photoelectron spectroscopy (XPS). MIDFDS was shown to have a sustained-release effect, with complete release near the cornea. Thanks to this dual-functioning carrier, a stable reduction in IOP for 10 h was achieved, and a promising clinical application of this system can be expected [65].

##### Other DDSs 

A dual DDS improving trabecular and uveoscleral outflow was introduced a thermosensitive hydrogel containing latanoprost and curcumin-loaded nanoparticles (CUR-NPs) in a study published by Cheng YH et al. in 2019. In addition to its effect on IOP, this hydrogel decreased oxidative stress-mediated damage in the TM [66].

In the study published by Donia et al. in 2020, the ocular delivery of carbonic anhydrase inhibitor acetazolamide (ACZ) in a stable suspension (NS) was researched. For its stabilization, two acids, namely anionic polypeptide poly γ-glutamic acid and the glycosaminoglycan hyaluronic acid, were used. A modified toxicity test proved the safeness and good tolerability in in vivo experiments on rabbit eyes; in addition, a hypotensive effect was detected [67].

In addition, silk-based biomaterials show great promise for ocular drug delivery in glaucoma treatment [68].

Natural biomaterials in DDSs are summarized in Table 2. Hybrid biomaterials used in DDSs are summarized in Table 3.

#### 3.3.2. Synthetic Biomaterials in DDSs

Brimonidine tartrate-loaded poly (lactic-co-glycolic acid) acid vitamin E-tocopheryl polyethylene glycol 1000 succinate (BRT-PLGA-TPGS) nanoparticles in a thermosensitive in situ gel were developed by Sharma et al. in 2021. This synthetic complex showed properties such as enhanced precorneal residence time with no eye irritation, both of which are important criteria for DDSs used in effective glaucoma management [70].

Cegielska et al. studied a nanofibrous DDS for brinzolamide delivery as an alternative to eye drops. A system based on natural β-cyclodextrin (β-CD), hydroxypropyl cellulose (HPC), and synthetic polycaprolactone (PCL) was designed. Each of the mentioned components is important, and each ensures a different function. HPC was responsible for the mucoadhesive and eye-hydrating properties, and PCL improved the mechanical performance of the carriers. Based on the results of this study, the developed system demonstrated all the necessary properties needed for ocular drug delivery in glaucoma treatment [69].

Dendrimer gel particles were developed and evaluated by Wang et al. in their study from 2021. The authors used polyamidoamine (PAMAM) dendrimers processed by a new method (the aza-Michael addition reaction method) to expand their capacity for drug encapsulation and delivery. Two first-line anti-glaucoma drugs were tested and loaded into dendrimer gel particles of different sizes. The authors concluded that this method could be clinically important in improving glaucoma treatment [71].

Jeong et al. reported a novel long-term Nitric Oxide (NO)-releasing polydiazeniumdiolate (NOP) that lowers IOP via the conventional outflow pathway. The monitored parameters in the study were real-time NO gas detection and release profiles. The safe level of NOP demonstrated TM relaxation and an IOP decreasing effect in vivo [72].

The potential of mesoporous silica nanoparticles loaded with sodium nitroprusside for the effective management of ocular hypertension was described by Hu et al. (2018) [73].

In a study by Song et al., brinzolamide-loaded core–shell nanoparticles made of a different inner part (poly-(DL-lactic acid-co-glycolic acid)) and outer part (2-diacyl-sn-glycero-3-phospho-L-serine (PS/PLGA]) were used to enhance the permeation of the drug during glaucoma treatment [74].

Synthetic biomaterials used in DDSs are summarized in Table 4. 

#### 3.3.3. Biomaterials in DDSs Used for Neuroprotective Treatment

Neuroprotective effects can be important as they can save retinal ganglion cells and the optic nerve head from damage by ischemia and oxidative stress. IOP is the main influenceable risk factor for disease progression, but it is well known that glaucoma damage can occur even in normotensive eyes. This suggests that factors other than IOP play a key role in its pathogenesis, and that neuroprotection can act as a therapeutic target. 

These different factors, such as mediators of apoptosis and lack of oxygen in tissues, can likely lead to disease progression; therefore, medication that could specifically target them is intensively studied. However, this field of research faces some challenges, including insufficient evidence for the effectiveness of these agents in clinical studies [75].

Arranz-Romera et al. studied neuroprotective medications in glaucoma using a DDS. The authors described and analyzed the incorporation of dexamethasone, melatonin, and coenzyme Q10 (D, M, Q) as a neuroprotective actors into multi-loaded PLGA-microspheres (MSs). In vivo efficacy studies were performed using a rodent model of chronic ocular hypertension (OHT) by comparing four groups: single intravitreal injections of DMQ-MSs, empty MSs, no OHT treatment (controls), and naive groups. Twenty-one days after OHT induction, DMQ-MSs showed a significant neuroprotective effect on retinal ganglion cells (RGCs) compared to the OHT controls [76]. The problem of neuroprotection and treatment with nanoparticles was also described by Silva et al. in 2022. Researchers have developed chitosan–hyaluronic acid nanoparticles (CS/HA) designed to carry epoetin beta (EPOβ) into the eye. The authors concluded that CS/HA nanoparticles could deliver EPOβ to the retina well; this complex was considered biologically safe and could have the potential to treat optic nerve degeneration associated with glaucoma [77]. 

## 4. Conclusions

In this study, novel biomaterials used for glaucoma treatment are reviewed. Biomaterials have, due to their origin, favorable properties like biocompatibility, biostability, non-toxicity, and easy accessibility, and they show generally promising potential for the management of glaucoma disease. Moreover, thanks to new methods processing biomaterials, improving their abilities, or expanding their use to, for example, electrospinning technology, it is possible to create novel structures imitating the natural drainage system of the eye or improve the delivery of the antiglaucoma drugs into the target area. 

Novel biomaterials have already been found in the ophthalmology field and are currently being intensively studied for both surgical and conservative glaucoma treatments, which is still challenging. 

Exhaustive research has focused on novel materials for glaucoma drainage devices because existing implants still have limitations and a high probability of failure, especially in the long term. Regarding possible complications, implants made of flexible rather than rigid materials, such as metal, seem to be, in our opinion, more promising and deserve further investigation. Solid materials can damage the endothelial layer of the cornea, as stated by, e.g., Garcia-Feijoo [78]. In this respect, for example, the nanofibers non-degradable PVDF implant described by Klapstova et al. could be promising, although subsequent in vivo studies are lacking [20]. Nanofibers also show promise in the Josyula study, where an in vivo study has already been performed with encouraging results [22].

Another area, which deserves further research, is the use of biomaterials in DDSs because they have the potential to eliminate many of the negative effects associated with commonly used topical treatments. 

As glaucoma is still a disease threatening the sight of the population, both areas of biomaterials research are desirable and it is essential to support further in vivo studies. To date, only a third of the studies on novel biomaterials for the treatment of glaucoma mentioned in this review have been conducted under in vivo conditions, which leaves much work to be conducted.

## Figures and Tables

**Table 2 biomedicines-12-00813-t002:** Natural biomaterials in DDSs.

Natural Biomaterials	Type of Biomaterial	Drug to Be Delivered	Author, Year	Type of Study
Chitosan (CH)-based/loaded	CH-Gelatin hydrogel	Timolol maleate	El-Feky, 2018 [49]	In vitro, In vivo
	Montmorillonite/CH nanoparticles	Betaxolol hydrochloride	Li, 2018 [53]	In vitro
	CH/hydroxyethyl cellulose-based ocular inserts	Dorzolamide	Franca, 2019 [58]	In vivo
	CH-based thermosensitive hydrogel	Timolol maleate	Pakzad, 2020 [50]	In vitro
	CH and ZM241385 hollow ceria nanoparticles	Pilocarpine	Luo, 2020 [55]	In vitro, In vivo
	CH-g-poly(N-isopropylacrylamide) (CN)	Pilocarpine	Luo, 2020 [56]	-
	Chitosan nanoparticles	-	Li, 2020 [61]	In vitro
	Chitosan-pectin mucoadhesive nanocapsules	Brinzolamide	Dubey, 2020 [59]	In vivo
	Niosomes coated with CH	Carteolol	Zafar, 2021 [60]	-
	Mucoadhesive polymeric inserts prepared using CH and chondroitin sulfate	Benzamidine	Cesar, 2021 [57]	-
	CH-coated bovine serum albumin nanoparticles	Tetrandrine	Radwan, 2022 [54]	Ex vivo
Montmorillonite (MO)-based/loaded	MO/BH complex encapsulated into Eudragit microspheres	Betaxolol hydrochloride	Tian, 2018 [63]	In vitro
	Montmorillonite-loaded solid lipid nanoparticles	Betaxolol hydrochloride	Liu, 2020 [62]	
	MIDFDS	Betaxolol hydrochloride	Liu, 2021 [65]	-
Others	Thermosensitive hydrogel with curcumin-loaded nanoparticles	Latanoprost	Cheng, 2019 [66]	-
	Silk-based biomaterials	-	Wani, 2022 [68]	-
	Nanofibrous drug delivery system- β-cyclodextrin	Brinzolamide	Cegielska, 2022 [69]	In vivo
	Nanofibrous drug delivery system- Hydroxypropyl cellulose	Brinzolamide	Cegielska, 2022 [69]	In vivo

**Table 3 biomedicines-12-00813-t003:** Hybrid biomaterials in DDSs.

Type of Biomaterial	Drug to Be Delivered	Author, Year	Type of Study
Nanosuspension (NS) stabilized by anionic polypeptide, poly-γ-glutamic acid (PG), and the glycosaminoglycan, hyaluronic acid	Acetazolamide	Donia, 2020 [67]	-

**Table 4 biomedicines-12-00813-t004:** Synthetic biomaterials in DDSs.

Type of Biomaterial	Drug to Be Delivered	Author, Year	Type of Study
Mesoporous silica nanoparticles	Sodium nitroprusside	Hu, 2018 [73]	-
Core–shell nanoparticles	Brinzolamide-loaded PS/PLGA	Song, 2020 [74]	In vitro
Carbon-bound polydiazeniumdiolate	NO	Jeong, 2020 [72]	In vivo
Poly(lactic-co-glycolic acid) acid vitamin E-tocopheryl polyethylene glycol 1000 succinate (BRT-PLGA-TPGS) nanoparticles	Brimonidine tartrate	Sharma, 2021 [70]	-
Polyamidoamine dendrimers	Brimonidine tartrate/Timolol maleate	Wang, 2021 [71]	Ex vivo
Nanofibrous drug delivery system with polycaprolactone	Brinzolamide	Cegielska, 2022 [69]	In vivo

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
