# Peer review of "Novel Biomaterials in Glaucoma Treatment"

_biomedicines, 2024, doi:10.3390/biomedicines12040813_

Round 1

Reviewer 1 Report

Comments and Suggestions for Authors

This article reviews the use of novel biomaterials for glaucoma treatment.  This is an interesting review, which is densely written. However, It would be better to provide figure to show how these novel biomaterials (such as iStent, Xen….) used in glaucoma drainage or how these biomaterials work on drug delivery systems (In vivo study). In addition. The authors should suggest which novel biomaterials are more promising and which deserves more investigations.  The conclusions should put forth the main message of the manuscript and future directions.

Reviewer 2 Report

Comments and Suggestions for Authors

The paper is well prepared and written, In this paper novel biomaterials used for glaucoma treatment are reviewed. As authors mentioned, novel biomaterials have already been found in the ophthalmology field and are currently being intensively studied for both surgical and conservative glaucoma treatments. In addition,  the use of biomaterials in DDSs is also been advocated because they have the potential to eliminate many of the negative effects associated with commonly used topical treatments. Both areas of research are promising.  However, up to now, only a third of the studies on novel biomaterials for the treatment of glaucoma mentioned in this review have been conducted under in vivo conditions, which leaves much work to be done.

The content of this review is quite of value for interested readers and the structure of the paper is also well organized.

Comments on the Quality of English Language

acceptable after editors minor modification

Author Response

Thank you very much for your review and comments. We hope that in case of need the minor english corrections by editor will be done.

Round 2

Reviewer 1 Report

Comments and Suggestions for Authors

no more comment